# Do bromeliads affect the arboreal ant communities on orange trees in northwestern Costa Rica?

**Beatrice Rost-Komiya**[1], **M. Alex Smith**[2], **Pierre Rogy**[1], **Diane S. Srivastava**[1]*

**1** Department of Zoology & Biodiversity Research Centre, University of British Columbia, Vancouver, Canada, **2** Department of Integrative Biology, University of Guelph, Guelph, Canada

* srivast@zoology.ubc.ca

**Data Availability Statement:** The data used in this study are archived at the Dryad Digital Repository [Rogy P, Hammill E, Srivastava DS (2019) Data from: Complex indirect effects of epiphytic

## Abstract

Ants often interact with other invertebrates as predators or mutualists. Epiphytic bromeliads provide nesting sites for ants, and could increase ant abundances in the tree canopy. We surveyed ants in the foliage of orange trees that either hosted bromeliads or did not. To determine if observed associations between bromeliads and tree ants were causal, we removed bromeliads from half of the trees, and resurveyed ants six weeks later. Our results show that bromeliad presence is correlated with higher ant abundances and different species of ants on orange trees during the dry season. This increase in ant abundance was driven primarily by *Solenopsis* ants, which were both numerous and found to facultatively nest in bromeliads. Bromeliad removal did not affect either ant abundance or composition, potentially because this manipulation coincided with the transition from dry to wet season. Other ant species were never encountered nesting in bromeliads, and the abundances of such ants on tree leaves were unaffected by bromeliad presence or removal. Considering the importance of ants in herbivore regulation, our findings suggest that bromeliads–through their association with ants–could indirectly be associated with biological control in agricultural systems.

## Introduction

Ants are a powerful ecological force, making up more than 10% of animal biomass of the earth [1] and up to 80% of animal biomass in the tropics [2,3]. They play a key role in plant pollination, soil perturbation, and regulation of crop-damaging insects [1] and often benefit plants by preying on herbivores. These effects are particularly significant in tropical regions, where ant removal in plants resulted in a decrease in plant fitness of 59%, in part due to the role of ants in preventing herbivore damage [4,5]. However, ants also have mutualistic interactions with herbivores, such as aphid-tending and butterfly larva-tending behaviours, which may or may not benefit plants. On the one hand, tending behaviours by ants may result in higher densities of aphids and butterfly larvae and consequently higher herbivory rates and/or higher rates of disease-spread by aphids [6–9]. On the other hand, studies have shown that the majority

bromeliads on the invertebrate food webs of their support tree. Dryad Digital Repository. https://doi.org/10.5061/dryad.b4c364r]. All amplified sequences and specimen photographs can be accessed at: dx.doi.org/10.5883/DS-ASANTBRO.

**Funding:** B.R-K. was supported by the Natural Sciences and Engineering Research Council of Canada - NSERC (http://www.nserc-crsng.gc.ca) Undergraduate Student Research Award (#527311), D.S.S and M.A.S. were supported by NSERC Discovery Grants. The funders had no role in study design, data collection and analysis, decision to publish, or preparation of the manuscript.

**Competing interests:** The authors have declared that no competing interests exists.

(73%) of ant-hemipteran associations enhance ant predation and harassment of more damaging herbivores and thus confer a net benefit on the host plant [6,10]. Understanding these relationships can have important implications for the management of tropical agroecology operations, especially citrus orchards where ants are one of the most abundant insect groups [11]. A meta-analysis of the effects of ants on citrus pests and their natural enemies demonstrated the complexity of these relationships [11]. While ants are effective in controlling non-honeydew producing citrus pest abundance, their presence can also result in an increase in the abundance of honeydew-producing pests. That being said, the potential for ants to positively affect honeydew producers depends strongly on the identity of the ant species; *Lasius grandis*– an ecologically and behaviourally dominant ant species in Mediterranean citrus orchards–, for example, suppresses natural enemies of honeydew-producing pests by up to 145% relative to other less abundant ant species [11–13].

Many tropical ants have positive associations with plants, ranging in strength from facultative to obligate [14,15]. Well-known obligate associations include acacia ants of the subgroup *Pseudomyrmex ferrugineus*, which are only able to nest in domatia of their *Vachellia* host plant [16,17], as well as Neotropical ant gardens formed by a symbiosis between specific ants and specific plants [18]. Many other ant-plant associations are facultative, in that ants use a range of plant species. For example, the majority of domatia-using ants are able to occupy a range of host plant species [19]. Ants can indirectly benefit their host plants by reducing herbivore abundance [20] or by forming and enriching soil around plant roots [21]. When the host plant is epiphytic, these benefits of ants may extend to the support tree. Epiphytes have been shown to be correlated with higher ant abundances on plantation trees, compared to trees without epiphytes [22]. Furthermore, the presence of epiphytes on trees has often been found to be associated with an increase in ant species richness [23,24]. In the Neotropics, ants have both facultative and obligate associations with epiphytic plants in the family Bromeliaceae, nesting in terraria formed by their outer axils and roots and foraging in their leaf baskets [25,26]. In this study we use experimental manipulations of bromeliads in orange trees to examine how bromeliads influence the ant community on orange tree leaves.

A previous study [27] showed that orange trees with bromeliads tend to have lower leaf damage, especially when the bromeliads contain ants; however, this pattern is so far only correlative. Thus, the goal of our research is to first establish whether orange trees that naturally differ in bromeliad presence also differ in their arboreal ants, and then to test if any observed association is causal by following ant response to bromeliad removals. To further investigate the causal mechanism, we ask whether ant species known to nest in bromeliads are the most affected by bromeliad removal. To study these correlative and causal effects of bromeliads on arboreal ants, we implemented three treatments: trees without bromeliads (hereafter "without-without"), trees with bromeliads (hereafter "with-with"), and trees whose bromeliads were removed after the first round of sampling (hereafter "with-removal"). We hypothesize that bromeliads primarily affect ants by providing nesting habitat, and, because ants nesting in bromeliads forage on the host orange tree, bromeliads will increase arboreal ant densities. Specifically, we predict that: (1) ant abundance on orange tree leaves will be greater on trees naturally with bromeliads (Fig 1A: trees "with-with" bromeliads have more ants than those "without-without"); (2) removal of bromeliads will decrease ant abundance (Fig 1A: "with-removal" trees originally have similar ants abundances to "with-with" trees, but following the manipulation have fewer ants); (3) removal of bromeliads will result in the species composition of the ant community on orange tree leaves resembling that on trees naturally without bromeliads (Fig 1C: the "with-removal" ant communities move in multivariate space from near the "with-with" communities to near the "without-without" communities); and (4) any effects of bromeliad presence or removal on the ant community will be primarily due to ant species found

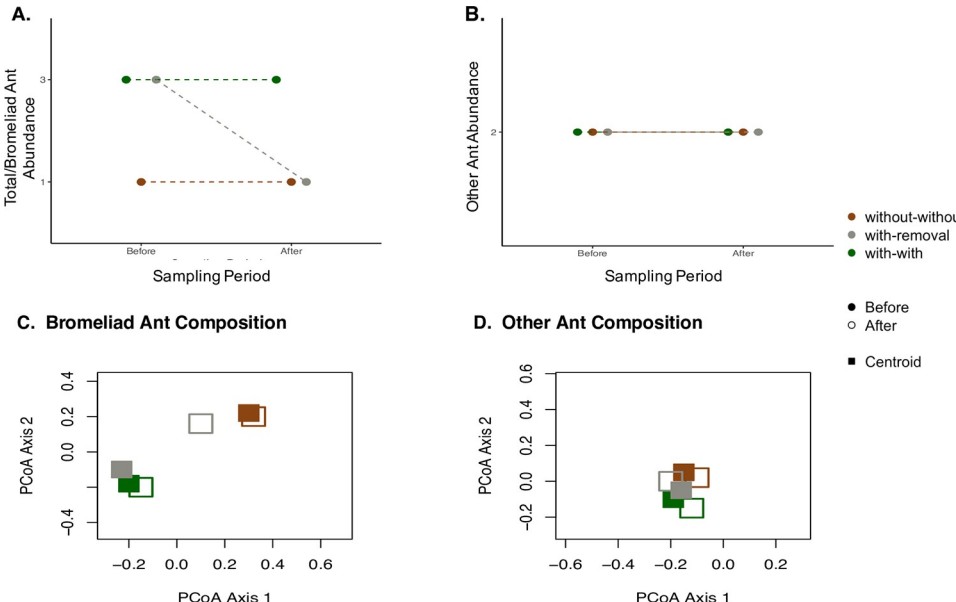

**Fig 1.** Figures depicting our predictions for A. Mean total ant abundance or "bromeliad" (i.e. species known to nest in bromeliads) ant abundance B. Mean "other" (not associated with bromeliads) ant abundance C. Principal coordinates analysis (PCoA) for bromeliad ants and D. PCoA for other ants on orange trees across both sites before and after bromeliad removal. Colours represent 3 treatments: Trees with bromeliads (with-with), trees without bromeliads (without-without), and trees with bromeliads removed (with-removal). Centroids represent these three fitted categories.

nesting in bromeliads ("bromeliad ants"), not "other" ant species (Contrast Fig 1A with 1B, and Fig 1C with 1D).

## Materials and methods

### Study sites

This study was conducted in two orange groves located around Santa Cecilia in northern Guanacaste, Costa Rica. This project was completed under Ministerio de Ambiente y Energía de Costa Rica (MINAE) research and collecting permits ACG-PI-012-2017, ACG-PI-PC-032-2017 and ACG-PI-PC-034-2017 and export permit DGVS-015-2017-ACG-PI-012-2017. Permission to DNA barcode the ants collected here was part of the Sistema Nacional de Areas de Conservacion (SINAC) approved collection of biodiversity resources for the BioAlfa project (CPI) No. CPI-SINAC-PNI-SE-001-2019. The operators of both sites agreed to preserve bromeliads on their trees to facilitate this study in November 2016. The two sites, hereafter CP and ER, are non-commercial groves of respectively 100 and 400 trees and were never subject to pesticide use. Moreover, both sites were surrounded by forest such that the distance of a given tree to any edge varied between 50m and 100m.

### Sampling design

For our experiment, we followed a Before-After-Control-Impact (BACI) design that consisted of two rounds of sampling: one pre-impact, and one post-impact [28]. This protocol allowed us to control for any pre-existing differences between replicates, as well as for any changes due solely to time. Because the BACI model inherently contains this dual control system, we can then attribute any variation we find in our results to our treatments.

In each of two study sites, we identified 20 trees naturally with and 20 trees naturally without bromeliads. Of these 20 trees with bromeliads, 10 trees would continue to bear bromeliads ("with-with") whereas the other 10 trees would eventually have their bromeliads removed ("with-removal"). Due to the shape of the tree canopies and the low height of the trees (2.5m to 4m), climbers were easily able to access bromeliads and visually ensure all large bromeliads were removed. Pre-impact sampling of all trees was conducted in May 2017 (hereafter "Before period"), before removing bromeliads from half of the trees that initially bore them. On average, trees contained about 3 bromeliads each. The arthropod community was allowed to recover from the disturbance for between 54–56 days and then post-impact sampling was conducted in June and July 2017 (hereafter "After" period). The duration of recovery was chosen based on the timeframes used by other similar studies [29]. The trees selected for sampling were devoid of domatia-providing epiphytes, and far enough apart that their crowns were not touching. Large bromeliads accounted for between one third and one half of the overall abundance of bromeliads on a given tree (longest leaf length $\geq$ 15cm). Because small bromeliads were extremely prevalent and found on the majority of trees in our study sites, trees bearing small bromeliads (longest leaf length < 15cm) were considered part of the bromeliad-free treatment. To minimize any potential effects these small bromeliads may have on the arthropod community, they were removed prior to sampling if empty of predatory arthropods. All fieldwork was led by P.R. as part of a larger, three-site study on how bromeliads affect arboreal insect food webs and leaf herbivory [30]. The third site, a commercial orange plantation, had such low ant abundances as to prevent analyses of ant composition, and is therefore excluded in this study.

## Field sampling

Within sites CP and ER, we set up four, three-dimensional sampling polygons (height: 50cm; width: 50cm; depth:100cm) directed towards the center each tree and sampled each polygon before (May 2017) and after (June-July 2017) the removal of bromeliads. In trees without bromeliads (without-without), the first polygon was randomly placed, whereas in trees with bromeliads in the Before period (with-with/with-removal), the first polygon was placed close to the tree's largest bromeliad (at a distance of one quarter of the tree's total circumference). The other three polygons were then placed clockwise, at regular intervals relative to the first polygon, around the circumference of the canopy. The same polygons were used for sampling in both the Before and After periods. To minimize disturbance due to our sampling procedure, we sequentially sampled eight trees per day. We collected ants and other arthropods by vacuuming continuously for a minute using a cordless, electric leaf vacuum (IONBV-XR Model, Snow Joe LLC, Carlstadt, NJ), identified collected ants to morphospecies, and preserved type specimens of each in 70% ethanol for more thorough identification at the University of British Columbia in Vancouver, Canada.

The 174 bromeliads that were removed were dissected leaf by leaf, and the presence of ant nests was inferred based on the presence of workers, eggs, and larvae within the detritus within a given bromeliad. On average, bromeliads measured 6.6cm ± 0.2cm in diameter with an average longest leaf length of 52.1cm ± 1.5cm. Nesting ants were preserved for later identification and these species are considered "bromeliad-associated" ant species for our analyses of the arboreal ant community.

Ant identifications were led by B.R-K and A.S. In the laboratory, we identified specimens collected in Costa Rica to subfamily, and–if possible–further to genus and species. To confirm provisional identifications, we point mounted individuals of each morphospecies, imaged them laterally using Leica Z16APO microscope and focus-stacked these images using the Leica

Microsystem Application Suite V4.3. In addition to morphological identifications, we used DNA barcodes to identify ant samples to a species level [28]. Briefly, an individual point side leg from each ant tissue was sampled for DNA extraction and the amplification of barcode region of the mitochondrial gene cytochrome *c* oxidase I (COI) using standard methods [28,31,32]. Successful amplifications were bi-directionally sequenced, aligned by eye and checked for insertions or deletions.

## Statistical analysis

We conducted our data analysis using the R programming language [33]). We categorized ants as "bromeliad-associated" if we found them at least once nesting in the dissected bromeliads, and as "other" ants if we did not find nests during the dissections.

To examine the effect of the bromeliad treatments on arboreal ant abundance at tree level, we fit generalized linear models with negative binomial errors to the ant abundance data using the glm.nb function in the MASS package [34], and analyzed the explanatory variables (site, treatment, and site x treatment) using an ANOVA.

To examine the effects of bromeliads on ant composition, we used the adonis function in the R vegan package [35] to run a PERMANOVA-type analysis [36] on the abundance data. The PERMANOVA analysis was conducted initially for all ant species, and then separately for "bromeliad-associated" ants and "other" ants to study the effect of the bromeliad treatments on ant composition. Dissimilarity was calculated using the Bray-Curtis method and 999 permutations. The explanatory variables in the PERMANOVA were "site" to test for differences between CP and ER, "treatment" to test for differences due to the presence, absence and removal of bromeliads, and "sampling period" to test for differences between pre-removal and post-removal sampling. We excluded any trees in CP and ER with abundance values of 0 because dissimilarity cannot be calculated between trees with no ant data. It is unlikely that male flying reproductives are trophically connected to the orange tree food web, whereas the queens are likely to be trophically connected due to new nest initiation; therefore, we chose to exclude alates from our statistical analysis.

To visualize any compositional changes occurring in the bromeliad-associated, and non-bromeliad ants between the pre-impact and post-impact sampling, we used principal coordinate analysis (PCoA), implemented in the vegan package. This involved generating a dissimilarity matrix using the vegdist function and producing an ordination using the cmdscale function. We fitted the treatment groups onto the ordination plot using the env.fit function.

We used an indicator species analysis to determine which species responded to treatments and could potentially provide evidence for the impacts of environmental change. The multipatt function in the indicspecies package [37] allowed us to generate indicator values for species based on individual species abundances and classification of the trees into their respective treatment groups. Because the multipatt can only generate a P-value for an indicator value if the species in question is strongly associated with a single of the three treatment groups, species like *Solenopsis* sp. A (BIN code: BOLD:AAF4712) that were dominant in all three treatment groups were not assigned a P-value. In such cases, we used the same mixed-effect ANOVA model from our earlier abundance analysis to determine whether there was a treatment effect for these individual species.

The data on ant community composition and species richness from this experiment has not been previously published. The data on ant abundance has been only presented before in aggregate form for a larger set of sites [30]. We re-analyse this abundance data just for the two sites in our study to compare patterns in total abundance to those of the other community descriptors.

## Results

### Abundance

Regardless of site, total ant abundance on orange tree leaves was driven by primarily by bromeliad-associated species, the ant species that we observed nesting in bromeliads at least once (Table 1, Fig 2). Total ant abundances were similar between the Before period, which coincided with the end of the dry season, and the After period, which was a few weeks after the start of the wet season (Fig 2A). Ant abundances were consistently higher at site ER than at site CP (Fig 2A). Furthermore, site ER also exhibited the most evident bromeliad effects on both the total ant abundances and bromeliad-associated abundances: trees originally with bromeliads (i.e., the with-with and with-removal treatments) had higher ant abundances than those originally without (without-without treatment) during both sampling periods (Table 1, Fig 2A and 2B). While the same pattern holds true at site CP in the Before period, it disappeared in the After period (Fig 2A and 2B).

In the Before period, we expected ant abundances to be similar on trees with bromeliads, regardless of whether these bromeliads were later maintained (with-with) or removed (with-removal) (Fig 1A and 1B). This was generally true (Fig 2). In the After period, we predicted bromeliad removal would reduce ant abundances in the with-removal trees to more closely resemble abundances in the without-without trees, particularly among the bromeliad-associated species, resulting in a significant sampling period x treatment interaction effect (Fig 1A). Our analysis, however, did not reveal significant sampling period x treatment effects within any of total ants, bromeliad-associated ants, or other ants (Table 1, Fig 2). While we were expecting the ant abundances on with-removal trees to change over time, it was the abundances on the without-without trees that changed between the Before and After periods (Fig 2). Specifically, a large increase in abundance by ants not usually associated with bromeliads seems to have driven an increase in total ant abundance on without-without trees (Fig 2C).

The most abundant ant species across both sites was the bromeliad-associated *Solenopsis* sp. A (BIN code: BOLD:AAF4712) (Fig 3A). Due to high abundance levels, *Solenopsis* sp. A was a key driver of the patterns we observed in both the bromeliad-associated ant abundances, as well as in the total ant abundances, particularly at site ER (Fig 2A and 2B). At site CP, other bromeliad-associated ants also contribute to the overall effects of bromeliads (Fig 2A and 2B). While *Solenopsis* sp. A was often found nesting in bromeliads, its presence on without-without trees suggests that the relationship between the two organisms may be more facultative than obligate (Fig 3A). This is supported by our field observations of this ant species nesting in the ground, as well as inside leaf rolls and bark crevices. In contrast to *Solenopsis* sp. A, the

**Table 1. Total ant, bromeliad ant, and other ant abundances modeled as functions of sampling period, and treatment.**

|  | All Ants | Bromeliad Ants | Non-Bromeliad Ants |
|---|---|---|---|
| **Sampling Period (SP)** | $X^2 (1) = 0.88$ <br> $P = 0.348$ | $X^2 (1) = 0.58$ <br> $P = 0.446$ | $X^2 (1) = 0.06$ <br> $P = 0.808$ |
| **Treatment (T)** | $X^2 (2) = 6.99$ <br> $P = 0.030$ * | $X^2 (2) = 16.42$ <br> $P < 0.001$ * | $X^2 (2) = 8.02$ <br> $P = 0.018$ * |
| **SP x T** | $X^2 (2) = 3.39$ <br> $P = 0.183$ | $X^2 (1) = 1.72$ <br> $P = 0.423$ | $X^2 (2) = 3.44$ <br> $P = 0.179$ |

All models are generalized linear models with negative binomial errors. Chi-square and P-values are generated by an analysis of variance (ANOVA).

"*" indicates statistical significance. Refer to Table 2 for a full list of ant species within each category.

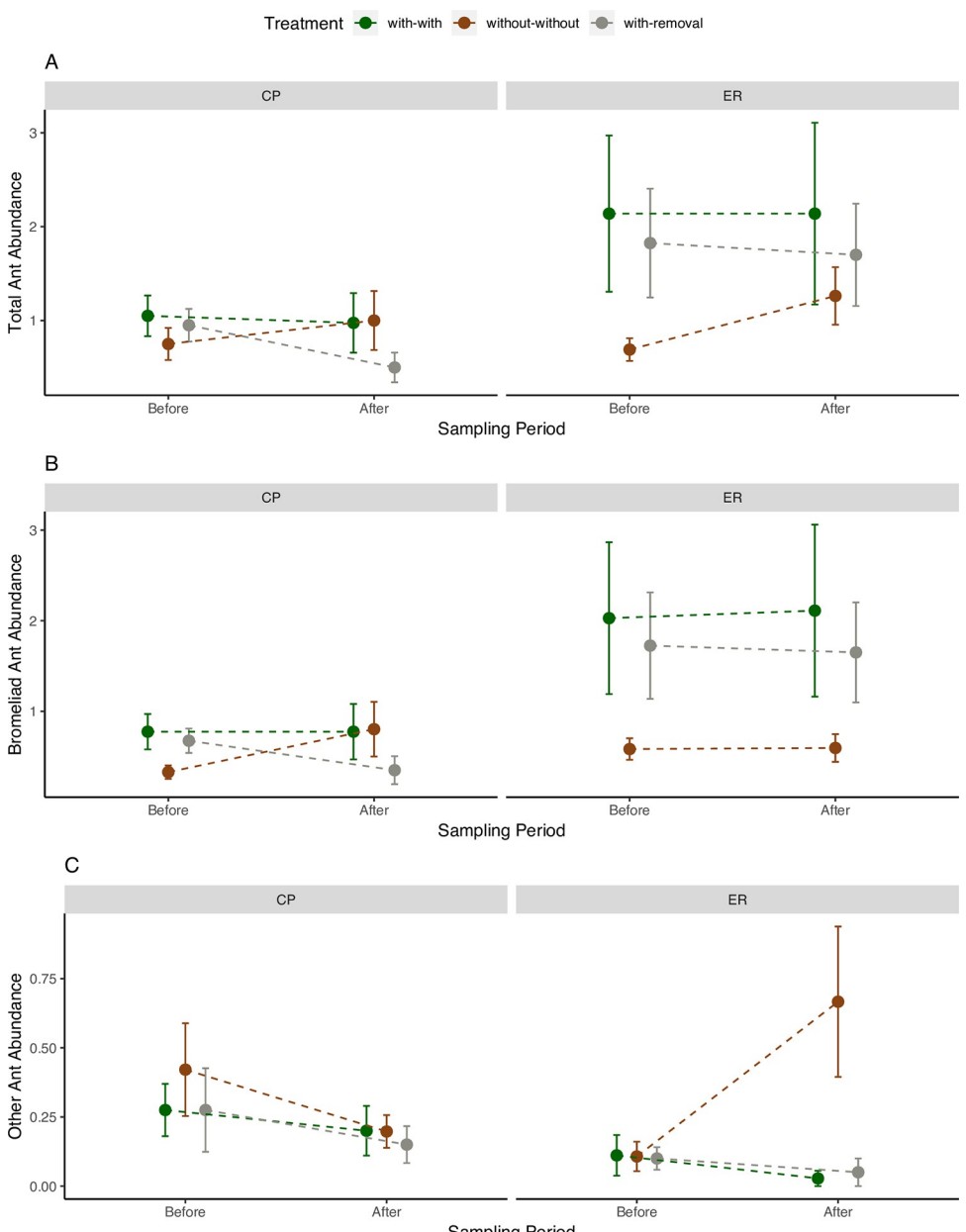

**Fig 2. Mean ant abundance (mean number of ants per 4 sampling polygons per tree ± SE) on orange trees at sites CP and ER before and after bromeliad removal.** A. Total ant abundance B. Bromeliad ant abundance C. Other ant abundance. Markers represent 3 treatments: trees with bromeliads (with-with), trees without bromeliads (without-without), and trees with bromeliads removed (with-removal) (n = 79).

indicator species analysis suggests that the relatively rare *Camponotus rectangularis* is more strongly associated with bromeliads, at least in the Before period (Table 2). At both sites, the highest abundances of this species were found in trees with bromeliads (with-with and with-removal) in the Before period, but abundances declined to near zero in the After period (Fig 3B). However, we caution that inference on this species is limited by small sample sizes: this species was found in only 9 trees, and nesting in bromeliads in only one of these trees.

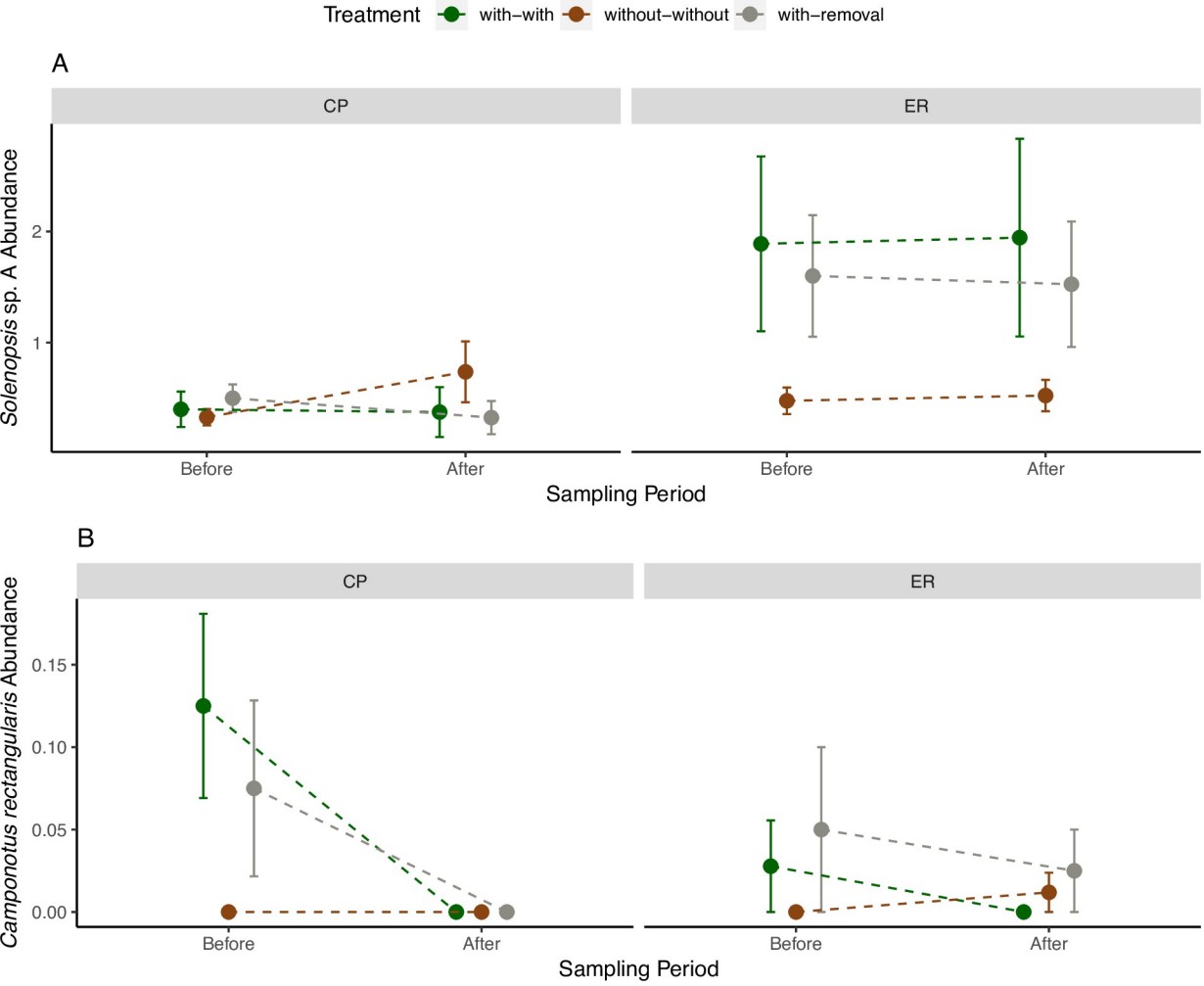

**Fig 3. Mean ant abundance (mean number of ants per 4 sampling polygons per tree ± SE) on orange trees at sites CP and ER before and after bromeliad removal.** A. *Solenopsis* sp. A abundance B. *Camponotus rectangularis* abundance. Markers represent 3 treatments: trees with bromeliads (with-with), trees without bromeliads (without-without), and trees with bromeliads removed (with-removal) (n = 79).

## Composition

We found that the presence of bromeliads on orange trees was associated with modest effects on the composition of the total ant community (40 species) (Table 3). The bromeliad treatments differed in the subset of the ant community that was bromeliad-associated (14 species), but not the subset of ants that were never found nesting in bromeliads (26 species) (Table 3). These results are in line with our prediction that the presence of bromeliads correlates with trees being populated by a different set of ant species than if bromeliads were absent, and our prediction that bromeliad nesting ants account for this effect. However, contrary to our prediction that the with-removal community would shift from resembling with-with in the Before period to more closely resembling without-without trees in the After period, we found no significant sampling period x treatment interaction for any of the three ant categories (Table 3, Fig 4).

## Discussion

We found that orange trees with bromeliads had, compared to trees without bromeliads, moderately different species composition of the complete ant community (40 species) (Table 3).

**Table 2. Indicator values and F values generated via ANOVA analysis for a) bromeliad-associated ant species and b) non-bromeliad-associated ant species.**

| a) Bromeliad-Associated Species | Sampling Period | | | | | | | |
|---|---|---|---|---|---|---|---|---|
| | Before | | | | After | | | |
| | Indicator Value | P | $X^2$ | P | Indicator Value | P | $X^2$ | P |
| Ant BD | – | – | – | – | – | – | – | – |
| Ant BF | – | – | – | – | – | – | – | – |
| Ant Z | – | – | – | – | – | – | – | – |
| *Azteca* MAS004 | 0.274 | 0.254 | – | – | – | – | – | – |
| *Camponotus atriceps* | 0.160 | 0.469 | – | – | – | – | – | – |
| *Camponotus rectangularis* | 0.453 | 0.004* | – | – | 0.183 | 1 | – | – |
| *Camponotus* sp. E | – | – | – | – | 0.3 | 0.275 | – | – |
| *Crematogaster brasiliensis* | – | – | – | – | 0.224 | 0.482 | – | – |
| *Nylanderia* MAS005 | 0.160 | 0.494 | – | – | – | – | – | – |
| *Nylanderia* sp. H | – | – | – | – | 0.3 | 0.275 | – | – |
| *Odontomachus* sp. AA | – | – | – | – | – | – | – | – |
| *Pheidole* ACJ4651 | – | – | – | – | 0.158 | 1 | – | – |
| *Pheidole* MAS027 | – | – | 14.26 | 0.278 | 0.158 | 1 | – | – |
| *Solenopsis* sp. A | – | – | 13.09 | 0.001* | – | – | 0.79 | 0.673 |

| b) Non-Bromeliad-Associated Species | Sampling Period | | | | | | | |
|---|---|---|---|---|---|---|---|---|
| | Before | | | | After | | | |
| | Indicator Value | P | $X^2$ | P | Indicator Value | P | $X^2$ | P |
| Ant AI | 0.158 | 1 | – | – | – | – | – | – |
| *Camponotus* sp. AF | 0.226 | 0.262 | – | – | 0.210 | 0.928 | – | – |
| *Camponotus* sp. AG | – | – | – | – | – | – | – | – |
| *Camponotus* sp. AM | 0.158 | 1 | – | – | 0.158 | 1 | – | – |
| *Camponotus conspicuus* | 0.224 | 0.5 | – | – | 0.158 | 1 | – | – |
| *Camponotus planatus* | – | – | 0.82 | 0.665 | 0.224 | 0.799 | – | – |
| *Cephalotes* sp. C | – | – | – | – | 0.203 | 1 | – | – |
| *Ectatomma* MAS001 | – | – | – | – | 0.224 | 0.524 | – | – |
| *Ectatomma* sp. Q | – | – | – | – | – | – | – | – |
| *Nesomyrmex* MAS001 | 0.224 | 0.492 | – | – | – | – | | |
| *Neoponera striatinodis* | – | – | – | – | 0.230 | 0.228 | – | – |
| *Nylanderia* MAS001 | 0.160 | 0.489 | – | – | – | – | 0.84 | 0.656 |
| *Paraponera clavata* | – | – | – | – | – | – | – | – |
| *Pheidole dossena* | – | – | – | – | 0.224 | 0.519 | – | – |
| *Pheidole pugnax* | – | – | – | – | – | – | – | – |
| *Pheidole punctatissima* | – | – | – | – | – | – | – | – |
| *Pheidole* sp. AW | – | – | – | – | 0.230 | 0.229 | – | – |
| *Pseudomyrmex simplex* | 0.160 | 0.481 | – | – | 0.158 | 1 | – | – |
| *Pseudomyrmex* sp. AD | – | – | – | – | – | – | – | – |
| *Pseudomyrmex* sp. AL | – | – | 0.86 | 0.650 | – | – | – | – |
| *Pseudomyrmex* sp. AO | 0.160 | 0.481 | – | – | 0.224 | 0.519 | – | – |
| *Pseudomyrmex* sp. O | – | – | – | – | 0.224 | 0.492 | – | – |
| *Pseudomyrmex* sp. P | 0.158 | 1 | – | – | 0.224 | 0.514 | – | – |
| *Pseudomyrmex* sp. Y | – | – | 2.25 | 0.324 | – | – | – | – |
| *Tapinoma* MAS003 | – | – | – | – | – | – | – | – |

*(Continued)*

**Table 2.** (Continued)

| | | | | | | | | |
|---|---|---|---|---|---|---|---|---|
| *Tapinoma* sp. AU | – | – | – | – | 0.400 | 0.121 | – | – |

Morphospecies that we were not able to associate with a taxonomic name are indicated with 'Ant' followed by a letter code. Blanks represent species whose abundances were too low to analyze or who were strongly associated with more than one of with-with, with-removal, and without-without trees.

"∗" indicates statistical significance.

This pattern was repeated in the subset of the ant community that was bromeliad-associated (14 species), but not the subset of ants that were never found nesting in bromeliads (26 species) (Table 3). These results are in line with our prediction that bromeliad presence is associated with a change in ant species composition, and our prediction that bromeliad nesting ants account for this effect. By contrast, there was little support for our prediction that removing bromeliads would shift ant communities from resembling those on "with-with" trees (Before period) to more closely resembling without-without trees in the After period. Specifically, we found no significant sampling period x treatment interaction for any of the three ant categories (Table 3, Fig 4).

Instead, the removal manipulation may have been ineffective because it coincided with the transition from dry (Before period) to wet (After period) seasons. If bromeliads benefit ants most in the dry season by providing cool, moist microhabitats, then removing bromeliads just before the wet season began may have had little impact on the ants [38]. The presence of epiphytes has previously been shown to contribute to an increase in insect individuals, particularly ants, of up to 90% [22,39]. These findings are consistent with the results from our dry season (Before) sampling period, during which bromeliads increased local ant abundance across both sites. During the wet season (After) sampling period; however, the effects of bromeliads on local ant abundance were site-dependent. We believe that seasonal fluctuations in insect abundance over the dry and wet seasons [40] coupled with the different degrees of association between particular ants and bromeliads, may explain why bromeliads seem to affect local ant abundance differently between the wet and dry seasons. Considering that ants are susceptible to desiccation, and that water-containing bromeliads provide habitats for potential prey [38], perhaps the increase in local ant abundance that we observed is due to bromeliad-associated ants' attraction to the water that remains stored in bromeliads during the dry season. In tropical ecosystems, seasonal increases in precipitation and humidity have been found to be associated with increased ant activity on tree trunks, as well as an overall increase in insect activity and local abundance [41]. Thus, during the wet season, increased prey

**Table 3. Total ant, bromeliad ant, and other (i.e. non-bromeliad) ant community composition modeled as a function of sampling period, treatment and their interaction.**

| | All Ants | Bromeliad Ants | Other Ants |
|---|---|---|---|
| **Sampling Period (SP)** | $F_{1,138} = 2.0$ <br> $P = 0.109$ | $F_{1,122} = 2.2$ <br> $P = 0.064$ | $F_{1,61} = 2.3$ <br> $P = 0.007$ ∗ |
| **Treatment (T)** | $F_{2,138} = 2.2$ <br> $P = 0.042$ ∗ | $F_{2,122} = 2.8$ <br> $P = 0.012$ ∗ | $F_{2,61} = 0.9$ <br> $P = 0.599$ |
| **SP x T** | $F_{2,138} = 0.9$ <br> $P = 0.470$ | $F_{2,122} = 1.2$ <br> $P = 0.323$ | $F_{2,61} = 0.9$ <br> $P = 0.569$ |

Community dissimilarity was calculated using the Bray-Curtis method. F- and P-values are generated by a PERMANOVA.

"∗" indicates statistical significance.

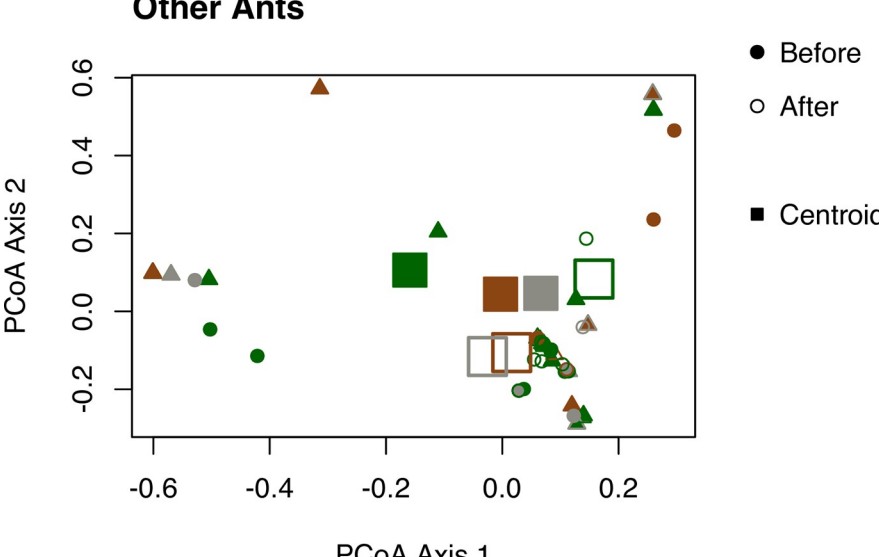

**Fig 4. Principal coordinates analysis (PCoA) for bromeliad ants (i.e. species known to nest in bromeliads), and other (non-bromeliad nesting) ants on orange trees at sites CP and ER before and after bromeliad removal.** Colours represent 3 treatments: trees with bromeliads (with-with), trees without bromeliads (without-without), and trees with bromeliads removed (with-removal). Centroids represent these three fitted categories.

availability and decreased risk of desiccation, may lead facultative bromeliad nesters to use other structures for nesting, consequently negating the positive effect of bromeliads on ant abundance.

The restriction of treatment effects to bromeliad-associated ants is consistent with our prediction that bromeliad nesters are the primary determinants of abundance patterns, as the other ants did not display the same pattern (Table 1, Fig 2). However, many of these nesting association with bromeliads are likely more facultative than obligate. For example, *Solenopsis*

sp. A was able to maintain populations on orange trees even after all bromeliads were removed, suggesting that it is also able to nest in other microhabitats on the trees. Similarly, *Camponotus rectangularis*–a species we found was strongly associated with bromeliads–has previously been shown to nest in bromeliads [42–44]; however, it is also known to establish mutualistic relationships with the orchid *Myrmecophila tibicinis* [45] and nest in various other tree species [44,46]. This indicates a degree of resilience of the populations of certain ant species to bromeliad removal. After observing similarly subtle effects of epiphytes on the composition of ant assemblages, Stuntz et al. [22] proposed that this may be due to a tendency for arboreal ants to behave highly opportunistically with respect to their host plants.

Epiphytic plants such as bromeliads have been shown to act as keystone species for the maintenance of ant communities and services [47]. Their architectural morphometry and the suspended soil formations associated with their roots are pivotal in structuring the forest canopy, providing vital nesting microhabitats, and offering foraging opportunities for a range of ant species [48]. The degree of association between Neotropical ants and bromeliads seems to vary depending on the species. In ant gardens, the workers of certain ant species collect and incorporate the seeds of tank bromeliads into their nests, and thus influence the shape and size of the bromeliad [49]. The nature of this particular relationship suggests a high degree of co-dependence between ant and bromeliad; however, the microhabitats provided by bromeliads also offer nesting and foraging opportunities for a range of more generalist canopy ants [50]. Our study provides further evidence that ant species found nesting in bromeliads account for much of the effect of bromeliads on the ant communities on tree leaves.

These results point to a potentially important role for bromeliads and their associated ants in tropical agroecosystems; however, the nature of their role may depend on the life history of the ant species, and on the biology of a given pest [11]. Ants have been shown to act as inducible defences against herbivory in mutualistic ant-plant systems such as the *Pseudomyrmex-Tachigali* system [5] and are recognized as having a significant role in the elimination of fungi and phytopathogens [51–53]. The presence of ants such as *Solenopsis* sp. A or *Camponotus rectangularis* that exhibit tending behaviours towards honeydew-producing may also have an indirect net positive effect on host plants by deterring non-honeydew producing herbivores [11]; however, these benefits have been found to decrease with leaf size in some systems [54] and be negated entirely by honeydew-producer herbivory in others [55]. In citrus orchards in particular, the potential of ants as biological control agents against crop pests is being actively explored, with results suggesting that ant presence is associated with increased abundance of honeydew-producing pests, but reduced abundance of pests that spend part of their lives in the soil. This reaffirms the importance of epiphytic plants like bromeliads in their role as a keystone resource in tropical systems [56], specifically as a correlate and potential determinant of ant community composition in the context of biological control.

## Supporting information

**S1 File. Inclusivity in global research questionnaire.**
(DOCX)

## Acknowledgments

PR thanks Edd Hammill for conceptual contributions to the experimental design. We would like to thank Calixto Moraga, Petrona Ríos, Ernesto Rodriguez, Del Oro S.A. (especially Hugo Segini) and the Area de Conservación Guanacaste for their invaluable help during this project.

All work was performed under a MINAE research permit. This is a publication of the Bromeliad Working Group.

## Author Contributions

**Conceptualization:** Beatrice Rost-Komiya, Pierre Rogy, Diane S. Srivastava.

**Data curation:** Beatrice Rost-Komiya, Pierre Rogy.

**Formal analysis:** Beatrice Rost-Komiya, M. Alex Smith, Pierre Rogy.

**Funding acquisition:** Beatrice Rost-Komiya, Diane S. Srivastava.

**Investigation:** Beatrice Rost-Komiya, Pierre Rogy.

**Methodology:** Beatrice Rost-Komiya, Pierre Rogy, Diane S. Srivastava.

**Project administration:** Beatrice Rost-Komiya, Pierre Rogy, Diane S. Srivastava.

**Supervision:** Pierre Rogy, Diane S. Srivastava.

**Visualization:** Beatrice Rost-Komiya.

**Writing – original draft:** Beatrice Rost-Komiya.

**Writing – review & editing:** Beatrice Rost-Komiya, M. Alex Smith, Pierre Rogy, Diane S. Srivastava.

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
