## [Decision Letter · Decision Letter 0]

25 Apr 2022

PONE-D-22-02509Do bromeliads affect the arboreal ant communities on orange trees in northwestern Costa Rica?PLOS ONE

Dear Dr. Diane S Srivastava

Thank you for submitting your manuscript to PLOS ONE. After careful consideration, we feel that it has merit but does not fully meet PLOS ONE’s publication criteria as it currently stands. Therefore, we invite you to submit a revised version of the manuscript that addresses the points raised during the review process.

We look forward to receiving your revised manuscript.

Kind regards,

Kleber Del-Claro, PhD

Academic Editor

PLOS ONE

Journal Requirements:

“This project was supported by NSERC Discovery Grants to DSS and MAS, and all work was performed under a MINAE research permit. This is a publication of the Bromeliad Working Group.”

“B.R-K. was supported by the Natural Sciences and Engineering Research Council of Canada - NSERC (http://www.nserc-crsng.gc.ca) Undergraduate Student Research Award (#527311), D.S.S and M.A.S. were supported by NSERC Discovery Grants. The funders had no role in study design, data collection and analysis, decision to publish, or preparation of the manuscript.”

Additional Editor Comments (if provided):

Dear Authors, I have revised your article following reviewer 1, considering the decline in peer review and time we need an answer for you. I agree with all the comments fo the reviewer that is an expert in this particular research field. I ask you to consider all of the suggestions and comments made for the reviewer. I personally liked a lot your study, recently we published a similar one* (will help you to answer the comment of L333 - reviewer 1) and others in the same systems in the last years. The arboreal ant community associated to bromelieads per se is an amazing system with the outcomes on agricultural systems, still more interesting. I studied Camponotus senex almost two decades ago in Brazil, that is also an ant with enormous potential to biological control in mangos plantation.

I am waiting for the new version of your paper. All the best,

Kleber

*Jr Pacheco, P.S.M. and Del-Claro, K. (2018), Pseudomyrmex concolor Smith (Formicidae: Pseudomyrmecinae) as induced biotic defence for host plant Tachigali myrmecophila Ducke (Fabaceae: Caesalpinioideae). Ecol Entomol, 43: 782-793. https://doi.org/10.1111/een.12665

Reviewers' comments:

Reviewer's Responses to Questions

**Comments to the Author**

1. Is the manuscript technically sound, and do the data support the conclusions?

Reviewer #1: Yes

2. Has the statistical analysis been performed appropriately and rigorously? 

Reviewer #1: Yes

3. Have the authors made all data underlying the findings in their manuscript fully available?

Reviewer #1: Yes

4. Is the manuscript presented in an intelligible fashion and written in standard English?

Reviewer #1: Yes

5. Review Comments to the Author

Reviewer #1: The manuscript “Do bromeliads affect the arboreal ant communities on orange trees in northwestern Costa Rica?” considers the effects of bromeliads on arboreal ant communities in orange crops. Despite sharing part of the data with the following manuscript (“Hammill, et al. 2014. Bromeliad‐associated reductions in host herbivory: do epiphytic bromeliads act as commensalists or mutualists?. Biotropica, 46(1), 78-82”), this study has a different approach being independent from the above. Therefore, this manuscript presents an original idea filling a gap that has arisen. The “Introduction” is well written and the “Methods” are clear but some sampling information is necessary. The statistical analysis sounds appropriate. My main concern is that the authors do not explore the fact that there are species that establish intimate relationships with bromeliads (Who are these species?) and may have great potential in biological control (L13-15). See Anjos et al. (2021) Ants affect citrus pests and their natural enemies in contrasting ways. Biological Control. https://doi.org/10.1016/j.biocontrol.2021.104611. These authors explore the potential of some ant species for biological control in citrus crops. I would like to see this issue further discussed in the “Discussion” section. Overall, I think the study is sound and I have only some minor comments.

Minor comments:

L20-21: See Rosumek et al. (2009). Ants on plants: a meta-analysis of the role of ants as plant biotic defenses. Oecologia, 160(3), 537-549.

L21-22: Ants also establish important relationships with some caterpillar species (Lycaenidae). Cushman et al. (1994). Assessing benefits to both participants in a lycaenid‐ant association. Ecology, 75(4), 1031-1041.

L22: See Styrsky & Eubanks (2007). Ecological consequences of interactions between ants and honeydew-producing insects. Proceedings of the Royal Society B: Biological Sciences, 274(1607), 151-164.

L23-24: and higher spread of disease by aphids.

L42: See Camargo et al. (2012). Natural history of the Neotropical arboreal ant, Odontomachus hastatus: nest sites, foraging schedule, and diet. Journal of insect science, 12(1).

L69: I liked Figure 1 representing the predictions.

L94: How many bromeliads were there (on average) on the trees? What is the average size of the bromeliads removed? The size and shape of bromeliads determines some attributes of the tree ant communities.

L95: How was the removal of the bromeliads? With a climber? How can the authors be sure that all the large bromeliads (> 15 cm) were removed if the trees are relatively large and have a complex canopy?

L96: How many bromeliads were there (on average) on the trees?

L97: Why this period? Is it enough time for the arthropod fauna to recover?

L100-103: What is the estimated prevalence of large bromeliads (> 15 cm) in orange groves?

L111: It is not a quadrant, but a cube (three dimensions), right? What was the structure of the cube?

L115: What distance?

L140-141: How did you look for nests or similar structures on the bromeliads? Did you locate alates, pupae and eggs in these structures?

L184: Table 1 does not show who these species are.

Figure 2. Does abundance have these low mean values?

L191. I suggest an improvement in the formatting of the tables in the manuscript.

L227. The authors believe that Camponotus rectangularis may be a specialist bromeliad-nesting species? Do you have information on the natural history of this species? See Camargo, R. X., Oliveira, P. S., & Muscedere, M. (2012). Natural history of the Neotropical arboreal ant, Odontomachus hastatus: nest sites, foraging schedule, and diet. Journal of insect science, 12(1).

L240. Table 2. What is Ant BD, Ant BF, Ant Z and Ant AI?

Figure 4. Isn't it too much information to put the centroids on these graphs?

I was curious to know how many ant species were sampled in total? And how many were considered bromeliad ants? I think this information should be in the text. Maybe I missed it, but I would also like to know who are the bromeliad ants…

Perhaps it would be interesting to have a table (e.g., Supplementary Material) with the relative abundance of each ant species in the treatments considered in this manuscript.

L271-282: This paragraph is a bit confusing!

L279: How far away from the forest were the selected trees? Was the edge effect not considered?

L286: Please, provide references.

L295-298: Isn't it a contradictory idea to link desiccation with an increase in ant abundance in the dry season?

L334: See Camargo et al. (2012). Natural history of the Neotropical arboreal ant, Odontomachus hastatus: nest sites, foraging schedule, and diet. Journal of insect science, 12(1).

6. PLOS authors have the option to publish the peer review history of their article (what does this mean?). If published, this will include your full peer review and any attached files.

Reviewer #1: No

---

## [Author Response · Author response to Decision Letter 0]

9 Jun 2022

Please see our cover letter and our response to reviewers letter, which cover respectively requests made by the journal and by the reviewers.

---

## [Editor Report · Decision Letter 1]

23 Jun 2022

Do bromeliads affect the arboreal ant communities on orange trees in northwestern Costa Rica?

PONE-D-22-02509R1

Dear Dr. Diane S Srivastava,

We’re pleased to inform you that your manuscript has been judged scientifically suitable for publication and will be formally accepted for publication once it meets all outstanding technical requirements.

Kind regards,

Kleber Del-Claro, PhD

Academic Editor

PLOS ONE
---

## [Editor Report · Acceptance letter]

27 Jun 2022

PONE-D-22-02509R1 

Do bromeliads affect the arboreal ant communities on orange trees in northwestern Costa Rica? 

Dear Dr. Srivastava:

I'm pleased to inform you that your manuscript has been deemed suitable for publication in PLOS ONE. Congratulations! Your manuscript is now with our production department. 

Kind regards, 

on behalf of

Dr. Kleber Del-Claro 

Academic Editor

PLOS ONE